# Network Analysis Provides Insight into Tomato Lipid Metabolism

**DOI:** 10.3390/metabo10040152

**Published:** 2020-04-14

**Authors:** Anastasiya Kuhalskaya, Micha Wijesingha Ahchige, Leonardo Perez de Souza, José Vallarino, Yariv Brotman, Saleh Alseekh

**Affiliations:** 1Max Planck Institute of Molecular Plant Physiology, 14476 Potsdam-Golm, Germany; Kuhalskaya@mpimp-golm.mpg.de (A.K.); Wijesingha@mpimp-golm.mpg.de (M.W.A.); LPerez@mpimp-golm.mpg.de (L.P.d.S.); Vallarino@mpimp-golm.mpg.de (J.V.); brotman@mpimp-golm.mpg.de (Y.B.); 2Department of Life Sciences, Ben Gurion University of the Negev, 84105 Beersheva, Israel; 3Centre of Plant Systems Biology and Biotechnology, 4000 Plovdiv, Bulgaria

**Keywords:** lipophilic compounds, lipid-related genes, lipid metabolism

## Abstract

Metabolic correlation networks have been used in several instances to obtain a deeper insight into the complexity of plant metabolism as a whole. In tomato (*Solanum lycopersicum*), metabolites have a major influence on taste and overall fruit quality traits. Previously a broad spectrum of metabolic and phenotypic traits has been described using a *Solanum pennellii* introgression-lines (ILs) population. To obtain insights into tomato fruit metabolism, we performed metabolic network analysis from existing data, covering a wide range of metabolic traits, including lipophilic and volatile compounds, for the first time. We provide a comprehensive fruit correlation network and show how primary, secondary, lipophilic, and volatile compounds connect to each other and how the individual metabolic classes are linked to yield-related phenotypic traits. Results revealed a high connectivity within and between different classes of lipophilic compounds, as well as between lipophilic and secondary metabolites. We focused on lipid metabolism and generated a gene-expression network with lipophilic metabolites to identify new putative lipid-related genes. Metabolite–transcript correlation analysis revealed key putative genes involved in lipid biosynthesis pathways. The overall results will help to deepen our understanding of tomato metabolism and provide candidate genes for transgenic approaches toward improving nutritional qualities in tomato.

## 1. Introduction

Plants produce a wide variety of biochemical compounds, starting at the central or primary metabolism which generates compounds absolutely vital for plant survival and continuing through the pathways of specialized or secondary metabolism [1,2,3,4,5]. Specialized metabolites display a tremendous diversity, are often specific to certain plant lineages, and play many different roles in adaptation to the environment [6]. Volatile organic compounds (VOCs) are often discussed as a subgroup of secondary metabolites, with their low molecular weight enabling movement across cell membranes and release into the surrounding environment [7]. Secondary metabolites can act as direct or indirect defense agents by deterring herbivores, fending off pathogens, and/or attracting predators or pollinators [8]. When it comes to human consumption of plants, secondary metabolites also fulfill an important role, since they can have beneficial health effects and some of them contribute to flavor [9,10].

Since lipids may either be primary (e.g., glycerolipids, phospholipids) or secondary (derived from the isoprenoid pathway) metabolites, they are often discussed in the literature separately from other metabolites [11,12,13]. Lipids fulfill many different functions, ranging from carbon storage via cell-membrane components to signaling molecules [14].

The cultivated tomato (*Solanum lycopersicum*) has been widely used for metabolomic studies [15]. With production of over 150 million metric tonnes in 2017, tomato is the second most consumed vegetable in Europe after potato and the first by market value (www.fao.org/faostat/en/#home). The drought-tolerant green-fruited relative, *Solanum pennellii*, has been successfully hybridized with the cultivated tomato, and the offspring from that cross has been used to identify a wide range of phenotypic and metabolic quantitative trait loci (QTL) [16,17,18]. The importance of tomatoes for human diet, combined with the availability of genetic diversity from wild relative species, makes tomato an optimal model crop for studying different aspects of plant physiology [19]. In recent years, tomato metabolism has been intensively studied. Tomato’s specialized metabolites in particular have received much attention, since many compounds are known to have positive health effects [20], serving among others as antioxidants. Many efforts have been undertaken to understand the genetic basis of their biosynthesis and to increase the production of these compounds in fruits [21,22,23,24,25,26].

However, due to its immense complexity and interconnection, it is a great challenge to understand plant metabolism as a whole. QTL mapping is a commonly used approach to dissect plant metabolism and identify genes in the corresponding pathways [27]. Although QTL mapping is useful for the elucidation of individual pathways, this approach is still not sufficient for dissecting metabolism in its entirety due to the latter’s complexity [28]. Thus, metabolite correlation network analysis has been suggested as an additional method for elucidating novel connections in plant metabolism [29]. Several studies have used a network approach to display the correlation between metabolic compounds. Schauer et al. (2006) combined mQTL and network analysis to elucidate the relationship between metabolic and yield-related traits over two seasons of a tomato introgression-line (IL) population [30]. The study revealed a modular network with intra-modular connections of amino acids, sugars, and phenotypic traits, and highlighted the connections between metabolic and phenotypic traits. A similar approach was used to obtain several novel insights concerning the interrelation of seed primary [31] and fruit secondary metabolites with yield-associated traits [22]. Production of seed oils with desirable fatty-acid content raised the interest of oilseed breeders and biotechnologists as a result of the growing demand for vegetable oil [13,32,33,34,35]. Studies also focused on tomato seeds because their oil is an excellent source of important fatty acids involved in plant growth [36,37]. Following research has also been extended to other tissues, e.g., tomato leaves [31,38].

Despite the large-scale tomato cultivation and the number of studies related to the fruit’s chemical composition and its quality, the majority of tomato metabolome research has focused on polar (non-lipid) primary and secondary compounds: sugars and their derivatives, amino acids, and organic acids. Traits such as aroma, color, and nutritional values, all deriving from secondary metabolites, have been extensively investigated, too [1,39]. However, lipid metabolism is still poorly studied in tomato, with several investigations focusing on cuticular waxes [40,41,42,43] and the characterization of genes involved in fatty-acid production [44].

In this study we performed a global correlation-based network analysis of metabolic traits, including lipids, volatiles, primary, and secondary metabolites from tomato fruit pericarp, and in addition plant phenotypic traits across a *S. pennellii* population in three field studies. This allowed us to expose the connectivity network between lipids and other branches of metabolism and key agronomical traits. We further generated a correlation network between the levels of lipophilic compounds and the expression of 1431 lipid-related genes in fruit and leaf. We were able to identify 123 potential lipid-related genes and obtained several novel insights concerning lipid metabolism.

## 2. Results

### 2.1. The Overall Metabolic Network in S. pennellii ILs

*S. pennellii* ILs have contributed over the years to the investigation of more than 2500 QTLs associated with plant morphology and to our understanding of the genetic regulation of primary, secondary metabolism, and lipids [31]. Here, in order to decipher the underlying regulatory organization of lipid metabolism together with other metabolic traits, we created a correlation-based network using already existing data of primary metabolites [45], secondary metabolites [22], lipophilic compounds [44], volatile compounds [10], and yield-related traits [30] across *S. pennellii* ILs in three independent seasons: 2001, 2003, and 2004 (see Section 4, Appendix A). We correlated all metabolic and phenotypic traits against each other using Spearman’s rank correlation coefficient. Figure 1 shows an overall view of a correlation-based network of all collected data in *S. pennellii* ILs. As expected, high correlations occur more often within the same group of metabolites or phenotypic traits than between different groups. The vast majority of overall correlations in the network were positive (84.4%) (Appendix A).

### 2.2. Lipophilic Compounds Expose Weak Correlation with Phenotypic Traits and Primary Metabolites

We initially focused on correlations between lipids and phenotypic traits, volatiles, primary, and secondary metabolites (see Section 4).

Few correlations between different lipid classes and yield-related traits such as brix and plant weight were identified (Figure 2a). Specifically, galactolipids were correlated negatively to plant weight. Our results showed correlations between phenotypic traits and other metabolic classes, with 46% of traits being connected to primary metabolites. Brix revealed more connections to primary metabolites (primarily sugars) than to phenotypic traits. Brix reflects the total amount of soluble solids and therefore cannot be truly regarded as a phenotypic trait.

The network showed negative and positive correlations among phenotypic traits and various volatile compounds (Figure 2b). Lipid-, amino-acid-, and carotenoid-derived volatiles showed 69.2%, 55%, and 90% positive correlations with several phenotypic traits, respectively. Lipid-derived volatile compounds (hexanal, trans-2-hexenal, cis-3-hexen-1-ol) showed some positive correlations with flower-, seed-, and yield-related traits. In addition, yield-related traits showed a high number of negative correlations with amino-acid-derived volatile isovaleronitrile. Two secondary metabolites (calystegine A_3_ and calystegine B_2_) showed several correlations with phenotypic traits (Appendix A).

Our data show strong, mainly positive (92.1%) correlations within the class of primary metabolites. In agreement with previous studies, we observed a highly interconnected amino-acid module (Gly, Ile, Val, Thr, and Ser) [31]. However, primary metabolites such as galactinol and guanidine highly connected with specialized metabolites. In addition, two alkaloids (calystegine A3 and calystegine B2), belonging to the class of secondary metabolites, presented a high number of correlations with primary metabolites. 

We observed seven links between primary metabolites and lipophilic compounds such as phospholipids, glycerolipids, and galactolipids. 

Analysis of primary metabolites against volatile compounds indicated no significant correlations between amino-acid-derived volatiles like isovaleronitrile, benzaldehyde, 3-methylbutanal, and 2-phenylethanol and their upstream biosynthesis substrates such as leucine and phenylalanine. However, guanidine showed correlations with all amino-acid-derived volatiles. In total, amino-acid-derived volatiles revealed 80.8% positive correlations with primary metabolites. The majority of correlations between lipid-derived volatiles (trans-2-hexenal, hexanal, cis-3-hexen-1-ol) and carotenoid-derived volatiles (geranylacetone and b-ionone) with primary metabolites were positive.

### 2.3. Lipids Are Highly Correlated with Specialized Metabolites and Show Some Negative Correlations with Volatiles

Our results showed a high number of correlations between various lipophilic compounds and secondary metabolites, with 4588 correlations representing 30.8% of all edges shown in the network (Appendix A). Our data introduced for the first time, a high correlation between specialized metabolites and lipophilic compounds. For example, phospholipids were correlated to almost all secondary-metabolite subclasses. Glycerolipids showed 1725 positive and 335 negative correlations with secondary metabolites. A high number of positive correlations (96%) was also found between galactolipids and secondary metabolites (Table 1, Appendix A).

We identified 3249 correlations among various groups of secondary metabolites. Only 2% of all identified correlations were negative. Results indicated a high connection between and within different specialized-compound subclasses.

Correlation analysis revealed 10 positive and five negative interactions between lipid-derived volatiles and secondary metabolites. Correlations between amino-acid-derived volatiles and secondary metabolites showed an almost equal amount of positive and negative connections. Our observations revealed 58.7% (27 out of 46) of negative connections between carotenoid-derived volatiles and specialized metabolites (Figure 3a, Appendix A). Additionally, a high number of negative correlations between triacylglycerols and volatiles was observed. Other lipid classes were connected to volatiles mostly positively (Figure 3b, Appendix A).

Within all lipid classes we identified 5841 correlations, representing 39.2% of all edges shown in the network, with 1012 (17.3%) being negative. Lipids belonging to the same subclass showed strong, mainly positive connections with other members of the same class (Table 2). For example, among different triacylglycerols (TAGs), out of 1229 identified correlations 1102 (89.7%) were positive. There was also a minor number of negative correlations within the monogalactosyldiacylglycerol (MGDG) and digalactosyldiacylglycerol (DGDG) subclasses. Out of all identified correlations within the phospholipids subclass, 78.1% were positive. Moreover, there are no negative correlations between diacylglycerols (DAGs) and DGDGs. The highest percentage of negative correlations was observed between phospholipids and TAGs (Table 2, Appendix A).

### 2.4. Fruit-Specific Lipid-Related Genes Show a Mainly Positive Pattern of Change with Lipophilic Compounds

For further exploration of lipid biosynthesis in tomato fruit pericarp, we correlated lipid profiles across 76 introgression lines [44] with multiple next-generation-sequencing gene-expression datasets from the same introgression lines. We used data consisting of 188 and 117 different lipophilic compounds from leaves and fruits of 74 ILs, respectively, and gene-expression data of lipid-related genes (1431) to perform the correlation analysis.

Using Spearman rank correlation coefficient, we identified 59 positive connections out of 173 in fruits and 33 positive correlations out of 73 in leaves. Figure 4 shows the significant correlation values between gene-expression levels and different lipid concentrations in fruits (Figure 4a) and leaves (Figure 4b). 

Our analysis identified 123 potential lipid-related genes in both fruit and leaf datasets (Appendix A). In agreement with previous results [44], we observed that lipase (*Solyc12g055730)* showed a significant high correlation with different levels of TAGs (Figure 4a).

Our fruit data showed that the expression of the gene for 1-acyl-sn-glycerol-3-phosphate acyltransferase (GPAT) (*Solyc11g065890*) is positively correlated with 18 unsaturated TAGs, four of which showing significant mQTLs in IL 11-2 and IL 11-3 [44]. Further, expression analysis revealed that in IL 11-2 this gene is expressed 1.5 times lower compared to M82, and that in IL 11-3 expression of the gene is 0.69 times higher than in M82. Moreover, various TAGs display correlations with lipid-related genes putatively annotated as phospholipase D (*Solyc01g103910*)*,* non-specific lipid transfer protein (*Solyc10g075150*), acyl-ACP thioesterase (*Solyc12g006930*), and lipoxygenase (*Solyc03g122340*), whereas another lipoxygenase (*Solyc01g099210*) showed five positive correlations with various phospholipids. We also observed connections of galactolipids to several lipid-related genes (Appendix A).

Finally, we identified strong correlations between lipophilic compounds and the expression of a class III lipase (*Solyc09g091050*) located on chromosome 9. The gene was negatively correlated with DGDG 36:4 (–0.46). Our previous QTL data showed a significant mQTL for galactolipids and phospholipids in this region. The QTL was mapped to a narrow overlapping region of IL 9-3, IL 9-3-1, and IL 9-3-2. The levels of DAGs, DGDGs and MGDGs, and TAGs were significantly decreased in IL 9-3, IL 9-3-1, and IL 9-3-2 compared to M82. Furthermore, gene expression was 4.6-fold higher in cultivated tomato compared to introgression lines harboring the gene (Figure 5).

To provide additional support for the observed results, we performed genomic sequence analysis of the promoter region of the lipase (*Solyc09g091050*) and compared the promoter sequences of *S. lycopersicum cv. M82* and *S. pennellii* by genome alignment [17]. Results showed several small deletions and nucleotide substitutions in the promotor region, while the coding region showed 99% similarity between *M82* and *S. pennellii*.

Additionally, we investigated several significant correlations between different lipid classes and other lipid-related genes in tomato fruits (Appendix A).

### 2.5. Leaf-Specific Lipid-Related Genes Show Many Negative Correlations with Lipophilic Compounds

Similar to the above, we extended our analysis and combined leaf lipid profiling of ILs [44] with transcriptomic data from the same lines [46,47]. Using Spearman rank correlation coefficient, we identified 45.2% (33 out of 73) and 54.8% (40 out of 73) positive and negative correlations, respectively (Figure 4a). Our results show correlations between lipid transfer protein (*Solyc03g079880*), 3-ketoacyl CoA thiolase 1 (*Solyc09g061840*), and glycerophosphoryl diester phosphodiesterase (*Solyc11g045040*) with different DAGs and phospholipids. Phospholipids additionally were linked to phospholipid-translocating flippase (*Solyc01g011100*), diacylglycerol kinase (*Solyc01g096500*), and lipid transfer proteins (*Solyc03g119210* and *Solyc10g075070*). Moreover, one transfer protein, *Solyc03g119210*, was correlated with galactolipids (DGDG 32:3 and SQDG 32:1). Many other significant correlations between different lipid classes and lipid-related gene candidates were identified (Appendix A).

The cholesterol acyltransferase gene (*Solyc05g050710*), located on chromosome 5 (IL 5-3), predicted to take part in lipid catabolism, exhibited significant positive correlation with PC 32:0 (0.4). In the same region, several significant mQTLs for different lipid classes were identified [44]. In addition, the *Solyc05g050710* expression is almost 20-fold higher in *S. pennellii* compared to the cultivated variety (M82) (Figure 6a). Comparison of the coding sequence between *S. pennellii* and M82 revealed 99% identity. However, there are many differences in the promotor region, including a 17-bp deletion in *S. pennellii* compared to the cultivated variety, and additionally a 47-bp deletion in M82 compared to the allele derived from the wild species. This may account for the difference in the expression levels of *Solyc05g050710* between M82 and its wild relative *S. pennellii* (Figure 6b).

## 3. Discussion

### 3.1. Network Analysis: Correlation between Metabolic and Phenotypic Traits

Numerous efforts have been made to identify and characterize metabolic quantitative trait loci (mQTLs) in tomato, with a focus on primary and secondary metabolites [10,22,30,44,48,49]. In addition, QTLs for volatile organic compounds from tomato fruit [24,50,51,52] and acyl sugars in tomato leaf trichomes have been defined [53,54], with further studies focusing on natural variation [54,55,56,57,58] and cuticle composition [59]. A series of genome-wide association studies (GWAS) contributed to assessing the effects of domestication and crop improvement on the fruit metabolome [15,46,47,59,60,61,62]. Despite all these studies, relatively few investigations have hitherto aimed at understanding the genetic basis of lipid composition in tomato fruit, with few studies mainly focused on cuticular lipids [17,59,63,64].

Here, we investigated correlations between different classes of metabolites and phenotypic traits in fruits, combined with expression analysis of lipid-related genes in tomato fruits and leaves. The generated network partially validates previously discovered correlations and presents new ones. For example, a QTL for brix (*Brix9*-*2*-*5*) deriving from green-fruited tomato species increases glucose and fructose contents in cultivated tomato fruits [65]. We identified in our network significant positive correlations of brix with different sugars (Appendix A). Additionally, we observed strong correlations between amino acids (Gly, Ile, Val, Thr, and Ser), with an average *r*-value of 0.78 and small percentage of negative interactions of correlations within the class of secondary metabolites, in agreement with previous studies [22,31,65].

Large classes of lipophilic compounds, similarly as specialized metabolites, display a considerable diversity across different plant species, e.g., isoprenoid-derived compounds are considered “secondary” metabolites produced in a cell-specific manner and are not directly involved in cell growth and development [13]. Our results showed a high number of significant correlations (30.8% of all) between lipophilic and specialized compounds. The number of significant correlations between lipids and other metabolic traits is smaller.

Several yield-related traits showed correlations to galactolipids and to one phospholipid. Tomato fruit as an organ, unlike maize cobs, for example, does not store much lipids. Unlike seeds, tomato fruit and leaf cells do not accumulate high amounts of storage lipids. Lipophilic compounds in tomato fruits and leaves participating mainly in signaling, membrane structure, and development [66,67,68,69].

Furthermore, lipids, similarly to secondary metabolites, were less correlated with phenotypic traits (Figure 1). It has been shown experimentally that the variability in secondary metabolites does not impact morphological and yield-related traits [21,70,71]. Therefore, the ability to change lipid composition or levels in tomato fruits would be a valuable tool for improving fruit quality and flavor. For example, the composition of fatty acids can be significantly changed without altering the overall plant morphology [72]. Nevertheless, versatility of lipophilic compounds might indirectly affect traits like shelf-life, if cuticle lipids are changed [73]. Until now, the influence of lipophilic compounds on overall plant phenotype remains unclear.

The connection between secondary metabolites and lipids seems to be more direct since changes in one compound class can have an effect on the other. Experimental evidence has already highlighted the close connection between secondary metabolites and cuticle lipids [42], however, no experimental validation exists of the relationship between those two metabolic classes in fruit pericarp.

In our data, we identified several negative correlations between lipid-derived volatiles and various TAGs. The observations here confirmed previous finding in tomato fruit describing linkage between decreasing levels of TAGs and simultaneously increasing levels of volatiles originating from lipids [44]. In tomato fruit some volatiles are linked to overall liking and flavor intensity [25].

In our network, we observed strong correlations within the lipid class. Lipophilic compounds correlate mostly positively between each other. For example, most of the correlations between galactolipids and glycerolipids are positive. However, phospholipids showed numerous negative correlations with all other classes. These results indicate that phospholipids are functionally the most distinct lipid class. 

### 3.2. Network Analysis: Combining Metabolite Profiling and Expression for Gene Discovery

Plant lipid metabolism is constantly under investigation. In the well-studied model plant *A. thaliana* only around 40% of the 700 genes putatively annotated as lipid related are functionally characterized. In other plant species like tomato, that number is much smaller [74].

Recently we applied a quantitative genetic strategy to a *S. pennellii* IL population and mapped more than 160 various lipid species belonging to 10 different classes, with a total of 1528 and 428 mQTLs in fruit and leaf, respectively [44]. Here, we combined lipid profiling in leaf and fruit tissues across 76 ILs with gene expression analysis in order to identify genes involved in lipid biosynthesis.

To validate our approach, we checked whether our data confirmed previously proven correlations. We identified connections between a class III triacylglycerol lipase (*Solyc12g055730*) and various TAGs (TAG 48:2, TAG 58:0, TAG 48:3). It has been suggested that the enzyme catalyzes TAGs for further volatiles biosynthesis [44].

Our results highlighted several other lipid-related candidate genes in fruits and leaves (Figure 4). In tomato fruit, for example, the eQTL of lipase (*Solyc09g091050*) and mQTL of DGDGs and phospholipids confirmed high correlation between the gene and DGDG 36:4 (–0.46). Further validation of the gene’s function is required.

Furthermore, we identified a high number of lipid-related genes that correlated positively with TAGs. For example, the gene putatively annotated as acyl-ACP thioesterase (*Solyc12g006930*) correlates positively with nine TAGs with an average *r*-value of 0.43. The enzyme is essential in the process of chain termination during de novo fatty-acid synthesis [75]. Another example is the 1-acyl-sn-glycerol-3-phosphate acyltransferase gene (GPAT) (*Solyc11g065890*), which correlated with 18 unsaturated TAGs. The *A. thaliana* ortholog (*At3g57650*) was shown to be involved in phospholipid and TAG biosynthesis [76,77]. In tomato, GPAT catalyzes acylation at the *sn*-1 position of glycerol-3-phosphate to produce lysophosphatidic acid (LPA) with subsequent TAG synthesis [78]. For the same gene we found a correlation between level of expression from leaf dataset and MGDG 32:6. Differences in connections depending on the tissue type could suggest altered function of the same gene in fruits and leaves.

Interestingly, our data revealed mainly positive correlations between expression of lipid-related genes and levels of lipophilic compounds in tomato fruit, whereas in tomato leaves these were mostly negative (Appendix A). This may suggest that the genes involved in biosynthesis and regulation of lipid metabolism are generally different between fruits and leaves. These results are supported by a higher number of identified mQTLs in fruit compared to leaves [44]. This may indicate that lipid-related genes were less affected in leaves than in fruits in the context of the domestication process [79]. This could further mean that tomato fruits as sink tissues, which are dependent on carbon supply from source tissues, might need a tighter regulation of lipid production, compared to tissues like leaves where carbon is assimilated, making a flux to lipid metabolism “shorter” and more flexible [80].

In our leaf dataset, a high number of lipid-related genes were found to be correlated mainly with phospholipids and galactolipids compared to other subclasses. Galactolipids, for instance, represent the most abundant lipid class in thylakoid membranes, organelles specifically in leaves [81]. For instance, in our study we identified correlations between phospholipids and phospholipid-translocating flippase (*Solyc01g011100*), diacylglycerol kinase (*Solyc01g096500*), and lipid transfer proteins (*Solyc03g119210*, *Solyc10g075070*). Moreover, *Solyc03g119210* correlates with galactolipids. Another lipid-related gene candidate expressed in tomato leaves—3-ketoacyl CoA thiolase 1 (*Solyc09g061840*)—exposes correlations with DAGs and phospholipids [82]. The gene ortholog in *A. thaliana* was suggested to be involved in fatty-acid beta-oxidation [83]. Several other lipid-related genes such as particle serine esterase (*Solyc04g077180*) [84], cyclopropane-fatty-acyl-phospholipid synthase (*Solyc04g056450*) [85], acyl-CoA-binding protein (*Solyc08g075690*) [86], and long-chain fatty alcohol dehydrogenase (*Solyc09g090350*) [87] showed correlations with various phospholipids, galactolipids, and glycerolipids (Appendix A).

It has been observed that lipid metabolism could be genetically regulated on intra-class and inter-class levels [44]. We have identified several examples of genes following the pattern of intra-class level regulation such as GPAT (*Solyc11g065890*), which correlates with 18 TAGs. In contrast, the pattern of inter-class regulation was followed by lipid transfer protein gene (*Solyc03g079880*) or 3-ketoacyl CoA thiolase 1 (*Solyc09g061840*), which contributes to regulation of two different lipid subclasses simultaneously.

In this study, we evaluated metabolic trait correlations and performed global analysis of trait associations across a *S. pennellii* IL population. This is by no means the first time that network analysis has been used for evaluation of the relationships between traits in wide genetic populations, with many previous examples in Arabidopsis, potato, and maize [88,89,90]. Besides, the analysis has been applied for a range of different traits and tissue types in tomato populations [30,31,45]. However, here we included for the first time three major different lipid classes and revealed several insights concerning the interrelation of traits from yield-associated traits, primary, and secondary metabolism, volatiles with lipids.

Our network using correlation between gene expression and metabolite levels combined with DNA sequence analysis highlighted several candidate genes putatively involved in lipid biosynthesis or regulation. The presented results complement previous studies regarding metabolic traits in a *S. pennellii* IL population [10,22,30,44,45] and can be used for expanding the knowledge of lipid metabolism in tomato.

## 4. Materials and Methods

### 4.1. Plant Material

Data used in this study were based on *S. pennellii* ILs. The *S. pennellii* IL population was created by replacement of marker-defined genetic regions of the wild species *S. pennellii* with homologous fragments of the cultivated tomato *S. lycopersicum* (M82), representing whole wild-genome coverage of *S. pennellii* [16].

We used already available data for primary and secondary metabolites, lipids, and phenotypic traits. The data were obtained using a population grown in the Western Galilee Experimental Station in Akko, Israel, in a completely randomized design with one plant per m^2^. The field was irrigated with 320 m^3^ of water per 1000 m^2^ of field area throughout the season. The harvest of fruit was done when 80%–100% of tomatoes were red [16]. All the data were obtained from peeled-off fruit pericarp. Primary and secondary metabolites were available for three independent seasons: 2001, 2003, and 2004 [30,45]. Lipids data were available for seasons 2001 and 2003 [44]. Yield-related traits were available for seasons 2001 and 2004 [30], and flower-, seed- and fruit-related phenotypic traits—for season 2004 [30,91].

The data for volatile compounds were obtained from an *S. pennellii* IL population [10]. All lines were grown in randomized, replicated plots in three different sites (Gainesville, Citra, and Live Oak, Florida) over the seasons of 2002 to 2004. Volatile data we focused on in our study were consistently available only for season 2003. We also used available data for volatiles of interest for season 2004. Plants were grown using standard commercial practices in raised plastic mulched beds. Fruits from all plants for each line were combined and analyzed as they reached the red ripe stage [10].

Transcriptomic data across *S. pennellii* ILs RNA-seq of 1431 lipid-related genes from young leaf [47] and fruit peeled-off pericarp were obtained under http://ted.bti.cornell.edu [46]. We selected all expressed lipid-related genes across the tomato genome (based on GO). We extracted 647 and 786 lipid-related genes from leaf and fruit datasets, respectively.

### 4.2. Correlation Analysis

All metabolite and transcript values used for correlation analysis correspond to the standard scores of the log transformed data. Spearman correlation matrices were calculated in R (R Development Core Team, 2010) using the *cor* function of the *stats* package (https://www.rdocumentation.org/packages/stats). For the trait/trait correlations we used a critical *p*-value of 0.05, since it is a commonly used threshold for statistical analysis. Correlation *p*-values were obtained by performing a permutation test based on Spearman correlations using the using the perm.cor.test function of *jmuOutlier* package (https://www.rdocumentation.org/packages/jmuOutlier/versions/2.2/topics/perm.cor.test), set for 20,000 permutations. To select the most meaningful correlations for the network analysis, arbitrary cut-offs were set to an absolute correlation coefficient higher than 0.3 and 0.4 for trait/trait and trait/transcript networks, respectively. We set up the cut-off for trait/trait correlations so that known correlations would be incorporated in our network. Approved significant correlation between brix and sugars were shown before [65]. In our network the average correlation between brix and sugars were 0.378. Additionally, we used a relatively relaxed correlation coefficient threshold of 0.3, because we were integrating data from different platforms. For trait/transcript cut-off 0.3 we reported in total 1537 correlations with lipid classes. To get more insight on some of these correlations we decided to use a stronger cut-off of correlation coefficient (≥0.40).

Depending on the measurement or dataset, we had different amounts of replicates for the different traits in the introgression lines. For GC-MS replicate number was between 1 and 11, for LC-MS between 1 and 12, for lipids between 3 and 4, for phenotypic traits between 1 and 12 and for volatiles always 1. The number of replicates for M82 wild type was usually much higher.

The metabolite/metabolite network plot was produced using Cytoscape version 3.6.1 with nodes representing different metabolites and phenotypic traits and edges representing pairwise correlation above the set threshold. All metabolites and transcripts exhibiting at least one pairwise correlation above the metabolite/transcript correlation network cut-off were selected to be represented in the heatmap produced using the *pheatmap* (https://CRAN.R-project.org/package=pheatmap) package in R.

### 4.3. Promoter Analysis

Promoter analysis of *Solyc05g050710* and *Solyc09g091050* was performed on the accessions *S. lycopersicum* (M82) and *S. pennellii* (LA0716). Alignment of the promoter region of *Solyc05g050710* and *Solyc09g091050* was done with CLUSTALW (http://www.genome.jp/tools/clustalw/).

### 4.4. Trait Classes Used for Correlations

Phenotypic traits were divided to flower traits (anther length, anther width, anther length/width ratio, ovary length, ovary width, ovary length/width ratio, style length, style width, style length/width ratio), seed traits (seed length, seed width, seed length/width ratio, seed weight, seed number per fruit, seed weight per fruit, seed number per plant, seed weight per plant, seed number per fruit unit, inflorescence, flowers per inflorescence, flowers per plant), fruit-related (fruit length, fruit width, fruit length/width ratio, fruit pericarp thickness, fruit length/pericarp thickness ratio, fruit width/pericarp thickness ratio, fruit locule number) and yield-related ((brix (BX), brix yield (BY), plant weight (PW), total yield (TY), harvest index (HI), biomass (BM), fruit number (FN), red fruit weight (RED), earliness (EA), mean fruit weight (FW)) subgroups. Specialized metabolites were divided to flavonoids, glycoalkaloids, phenolics, N-containing compounds, hydroxycinnamate derivatives, acyl sugars, polyamines, and unspecified compounds subgroups, lipophilic compounds to trialycglycerols, diacylglycerols, phospholipids, digalactoyldiacylglycerols, monogalactoyldiacylglycerols subgroups, and volatile compounds to carotenoid-, lipid-, and amino acid-derived subgroups. Volatile compounds were divided to carotenoid-, lipid-, and amino acid-derived subgroups.

## Figures and Tables

**Figure 1 metabolites-10-00152-f001:**
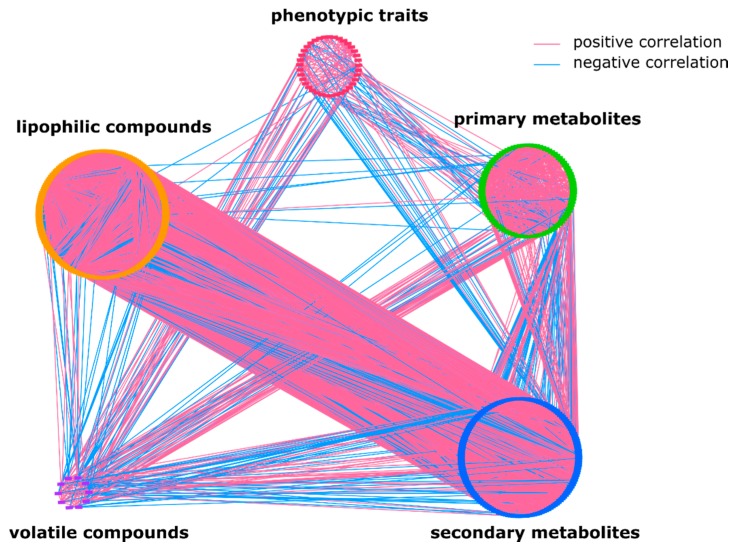
Overall metabolic network. Each node represents a metabolite or a whole plant phenotypic trait; edges connecting two nodes show an association between two traits. In total, the network is composed of 455 nodes and about 15,000 edges assembled into five large groups: lipophilic metabolites comprise of 171 nodes, primary and secondary metabolites have 89 and 147 nodes, respectively, phenotypic traits have 38 nodes, and the smallest group consists of 10 nodes and represents volatile compounds (Appendix A).

**Figure 2 metabolites-10-00152-f002:**
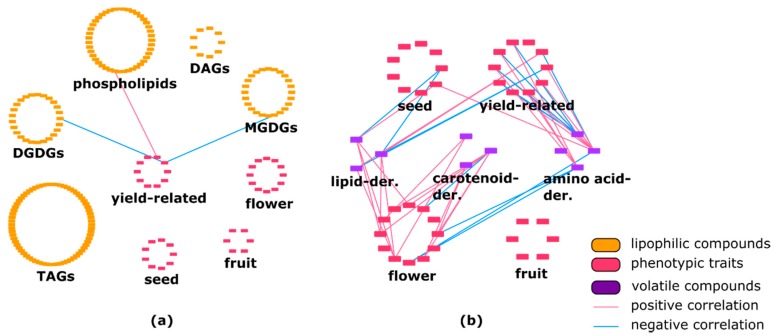
(**a**) Correlation between phenotypic traits and lipophilic compounds in *Solanum pennellii* introgression-lines (ILs). Yield-related traits showed three connection with several lipid classes such as phospho- and galactolipids. DAG—diacylglycerol; TAG—triacylglycerol; DGDG—digalactosyldiacylglycerol; MGDG—monogalactosyldiacylglycerol. (**b**) Correlation between phenotypic traits and volatiles in *S. pennellii* ILs. Various phenotypic traits are linked to each volatile class.

**Figure 3 metabolites-10-00152-f003:**
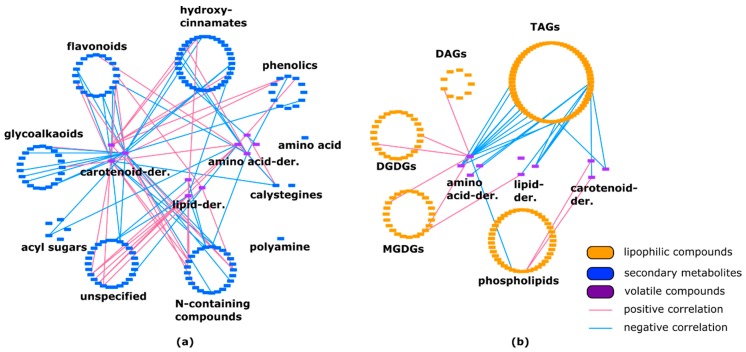
(**a**) Connection between secondary metabolites and volatiles in *Solanum pennellii* ILs. Volatile compounds are linked to almost all classes of specialized metabolites. (**b**) Correlation between lipophilic compounds and volatiles in *Solanum pennellii* ILs. Lipid class of TAGs showed numerous negative connections with all types of volatile compounds. DAG—diacylglycerol; TAG—triacylglycerol; DGDG—digalactosyldiacylglycerol; MGDG—monogalactosyldiacylglycerol.

**Figure 4 metabolites-10-00152-f004:**
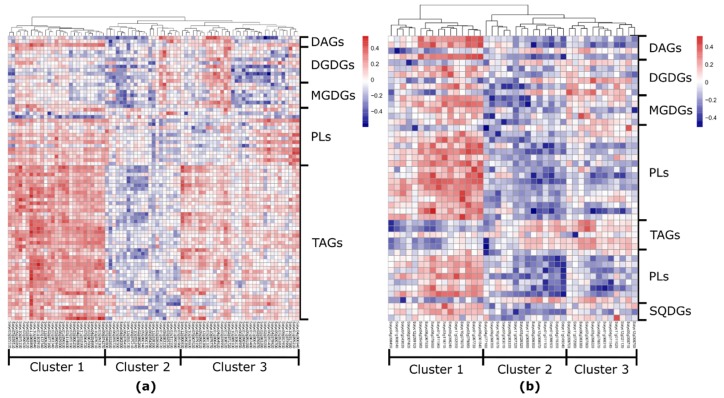
Heatmap of correlation values between lipid-related genes expression and lipid levels in tomato (**a**) fruits and (**b**) leaves. The number of lipid-related genes is higher in the fruit dataset compared to the leaf dataset (81 and 42, respectively). In tomato fruit, lipid-related genes linked mostly to TAGs, while in tomato leaf predominantly to phospho- and galactolipids. DAG—diacylglycerol; TAG—triacylglycerol; DGDG—digalactosyldiacylglycerol; MGDG—monogalactosyldiacylglycerol; SQDG—sulfoquinovosyldiacylglycerol.

**Figure 5 metabolites-10-00152-f005:**
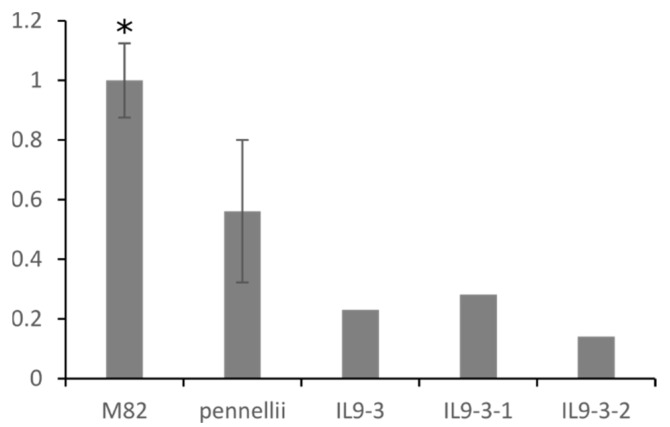
Transcript level of class III lipase (*Solyc09g091050*) in red ripe fruits of M82, *Solanum pennellii*, IL 9-3, IL 9-3-1, IL 9-3-2. Expression of the gene in wild tomato species *Solanum pennellii* as well as in ILs carrying the same allele version is lower in comparison to cultivated tomato variety M82. Asterisks indicate significant differences (* *p* < 0.05).

**Figure 6 metabolites-10-00152-f006:**
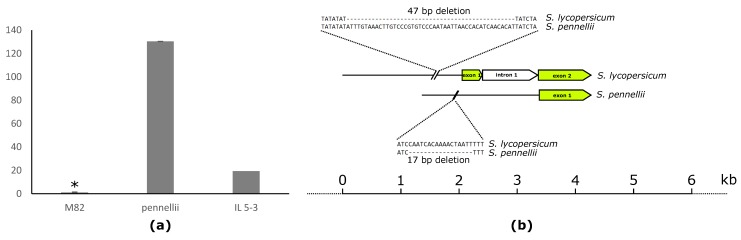
(**a**) Transcript levels of the cholesterol acyltransferase gene (*Solyc05g050710)* in red ripe fruits of M82, *Solanum pennellii*, IL5-3. Expression of the gene in wild tomato species *Solanum pennellii*, similar to IL 5-3 expressing the same allele version, is higher compared to cultivated tomato variety M82. (**b**) Comparison between the promoter regions of allelic versions of *Solyc05g050710* derived from cultivated tomato variety M82 and wild tomato *Solanum pennellii*. Deletions of 47 and 17 bp can be found in the promotor sequence of *Solanum lycorepsicum* and *Solanum pennellii* compared to the respective other. Asterisks indicate significant differences (* *p* < 0.001).

**Table 1 metabolites-10-00152-t001:** Percentage of positive correlations between lipids and secondary metabolites.

	MGDGs	DGDGs	Phospholipids	TAGs	DAGs
Acyl sugars	100	96	50	78	100
Flavonoids	99	93	51	88	97
Hydroxycinnamate derivatives	93	94	48	82	95
N-containing compounds	93	92	49	78	90
Unspecified	100	99	50	85	100
Amino acid	100	100	45	71	100
Glycoalkaloids	100	100	50	78	100
Others (phenolics)	98	98	53	79	100

**Table 2 metabolites-10-00152-t002:** Percentage of positive correlations between different lipid subclasses.

	MGDGs	DGDGs	Phospholipids	TAGs	DAGs
MGDGs	99	99	75	96	90
DGDGs	99	99	70	87	100
Phospholipids	75	70	78	60	65
TAGs	96	87	60	90	85
DAGs	90	100	65	85	100

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
