# Peer review of "Network Analysis Provides Insight into Tomato Lipid Metabolism"

_metabolites, 2020, doi:10.3390/metabo10040152_

Round 1
Reviewer 1 Report
Review Report
METABOLITES
Network analysis provides insight into tomato lipid 2 metabolism By Kuhalskaya et al.
Summary
The aim of this paper is to obtain insights into S. pennellii fruit metabolism by performing network analysis using published data from three field studies. The authors correlated metabolites (classified as primary, secondary, lipophilic, and volatile) and specific crop traits, namely plant weight, harvest index, and earliness, exposing the connectivity network between lipids and other branches of metabolism and key agronomical traits. Furthermore, a correlation network between lipids and the expression of lipid-related genes in fruit and leaf is presented.
Broad comments
The manuscript is too long, and that makes it confusing. I suggest that it be shortened by at least 20%. All sections can be shortened by removing content that is not fully relevant to the objectives and results of the manuscript, and by removing redundancy within and between sections. In several cases, statements can be made much more economically, i.e. with far fewer words. The Results section can be considerably shortened by removing discussion statements, and focusing on the outcomes and results of the experiments themselves. The Discussion should then be tightened, focussing on interpretation of these results.
The authors make particular emphasis on lipid metabolites, but do not include cuticle lipids -both waxes and cutin- which comprise an abundant lipid group produced by the fruit epidermis. However, genes highly represented in the transcriptomes used for the correlations presented here encode proteins of cuticle metabolism, such as an homolog of a GPAT acyltrasferase gene that is involved in cuticle precursor biosynthesis. Thus, correlations identified between cuticle biosynthetic genes and non-cuticular lipid components (i.e., TAGs) are difficult to interpret in terms of their metabolic/biological relevance.
Specific comments
Results
L24. Revise sentence.
L103. In addition to citing [10], [28], [36] and [38], cite Materials and Methods where the various traits are described.
L102,103. It is initially unclear if the authors measured these metabolites in S. pennellii ILs in three independent seasons (2001, 2003, and 2004), or used the metabolites published in the cited papers. The source of the data used should be clearly stated early on in the Results section.
L123. As mentioned by the authors, brix (total solid content) is not an agronomic trait. What is the value of using this parameter in the analysis?
L303-304. “… we extended our analysis and combined lipid profiling in leaves of ILs [37] with transcriptomic data from the same lines [56,57]. “ Are correlations meaningful if the transcripts and metabolites correspond to possibly very different growth conditions?
Discussion
L421. I agree that the cuticle, as a protective barrier, influences many plant traits. But these lipids have not been included in the analysis, so this discussion seems irrelevant. This also applies to comments on the cd2 mutant.
L461 and 464. Authors should mention this point (GPAT homolog) just once on the discussion.
Materials & Methods: I find the experimental work described quite confusing, it is uncertain what is new and what is taken from published work.
L541. For the metabolite data used in the study, the authors explain: “Data were retrieved for primary metabolites during three seasons 2001 and 2003 [26,38], secondary metabolites [28], lipophilic compounds [37] phenotypic traits [26,101] and volatile compounds available [10].” It is unclear to me how all these independent experiments are connected so that the correlations are valid. For example, in all these studies, were fruits harvested at the same time? Were all metabolites extracted from the same fruit tissues? What tissues, whole fruits? Same questions apply to the leaf data.
L569-574. Revise sentences.
L575. If the RNAseq data is not part of a published dataset, it should be properly described here. In addition, it is unclear what tomato tissues were used for this analysis. Whole fruits? Mixed leaves at various developmental stages? If the RNAseq is actually retrieved from the two sources cited by the authors, how do the tissues used to generate those transcriptomes relate to the samples used to study the different metabolite collections used for the correlations?
Reviewer 2 Report
The manuscript entitled “Network analysis provides insight into tomato lipid metabolism” by Kuhalskaya et al. describes a set of computational analyses performed on previously existing metabolomics and transcriptomics data from Solanum lycopersicum (tomato) introgression lines in combination with plant phenotypic data. The authors first examine relationships within and between groups of metabolites (primary, secondary, lipid, and volatiles) and phenotypic traits on a large scale (Fig. 1) and at a more detailed level (Figs. 2-4). They then combine the relationships they with further correlations they find between large-scale gene expression and lipid abundance (Fig. 5) and the expression levels and promoter sequences of specific genes (Figs. 6 and 7). All this information is integrated in the discussion to speculate on relationships among and between metabolites, transcript levels, and tomato phenotypes. Overall, the premise of the manuscript is interesting and is appropriate for Metabolites. However, if the manuscript is to be accepted for publication there are several major changes that need to be made to the text and perhaps to how data analysis was performed and certainly how it is explained. There are also many instances in which attention needs to be given to details of the manuscript before it can be considered for publication. Below are some suggestions for improving the manuscript.
Major aspects:
Data sources. The introduction, results, and methods sections need to be modified to make crystal clear where the data used was obtained and who performed that work. The methods section currently describes plant growth and harvest conditions, but then points to another paper (reference 18). Does this mean that the growth trial was first reported in reference 18 and that the data is now bring reused here? Or does this mean that a new growth trial was performed for the present study and that harvest was done using the methods reported in reference 18? If the growth trail was performed for this study then how was the data collected?
Data compatibility. This work heavily (exclusively?) relies on previously published data, which in itself is not a problem as the authors claim that it is being used in a new way to answer new questions. However, the issue of data compatibility needs to be explicitly addressed. The data used in this study are a collection of measurements from at least seven other publications. It is well established that environmental conditions exhibit major effects on plant metabolic, physiological, and transcriptional phenotypes. Can the authors be sure that no such variance is confounding their analyses?
Figures (and their captions) do not convey major points and/or are not linked to text. For example, for Figure 1, the first observation offered is that “high correlations” occur more often within the same group that between groups. However, in the figure it is impossible to distinguish lines that connect group members vs. lines connecting nodes from different groups. This comment applies to all the network figures in the paper. Consider using the facet feature of the R package ggnetwork as a means to fix this. Also, all figure captions are incomplete as they really only consist of a title and do not explain how to interpret the figure. This needs to be fixed throughout. In addition, consider, for example, lines 150-159. This text describes various results/findings, but no figures are ever referred to. How is the reader to follow along? Please fix this and any other instances of this issue throughout. Finally, in several places the authors compare numbers of positive vs. negative correlations between a set of nodes. For such comparisons, would bar charts summarizing the numbers of correlations not be better?
Methods are very limited. This leaves many questions unanswered, for example: why were cutoffs of 0.3 and 0.4 chosen? To the uninitiated reader, these seem very low. They also seem to make all networks extremely busy and difficult to interpret. The method by which these cutoffs were chosen needs to be explained to the reader. Methods also need to explain explicitly about who collected the data used in the analyses.
Discussion is very long. At the moment, the discussion is 194 lines (> 3 full pages) long and feels very long to read. There is considerable overlap with the results section, and several sections that do not appear to be driving at any particular conclusion or arguing a particular point. This entire section needs to be reworked so that it succinctly places the new results into the context of the literature and integrates them to make new inferences about plant biology.
Minor aspects:
Line 16 (abstract): “we performed extensive metabolic network analysis, covering a wide range…” this sentence should specify what data was used for the analysis (i.e. existing data) so as to keep reader expectations in check.
Line 19 (abstract): “how the individual metabolic classes influence yielded related phenotypic traits”. The word ‘influence’ in this context implies a causal relationship, which cannot actually be concluded from the results presented, which are truly only descriptive in nature. The word should be changed, and the manuscript should be searched for additional instances of this or similar errors.
Line 54: “wild relative species and therefore genetic diversity”. Is it always the case that phenotypic diversity is correlated with genetic diversity?
Line 62: “lacking comprehensive understanding”. Will we ever have a ‘comprehensive understanding’ of plant metabolism as a whole? i.e. is the goal implied by this sentence actually achievable?
Line 75: This sentence implies that oleic acid is an essential fatty acid, which it is not. Please fix.
Line 81: “Lipid metabolism is still poorly studied in tomato”. Do carotenoids not count as lipids? They have been extensively studied in tomato. If so, this sentence needs to be fixed so as to be accurate.
Line 93: if this work was not done as part of the present study, then a citation needs to be provided, or better yet, the sentence should be modified to make it absolutely clear when and by whom such work was done.
Line 136: “Showed only positive correlations”. This is inconsistent with the figure, since there are blue lines connecting lipid-derived volatile nodes with seed phenotype nodes. This needs to be fixed. Please check the entire manuscript for all similar claims of trends that are inconsistent with figures and correct accordingly.
Line 140: What does “EA” mean?
Line 142: “flower phenotypes being the most distinct subclass”. Are seed and fruit phenotypes not equally “distinct”? Please clarify.
Line 173: “cite xx”. Please fix.
Line 186: “phospholipids were correlated”. -> “phospholipids were negatively correlated”
Figure 3: (a) is so hard to read as many lines cover one another up. Is there not a better way to show these results? Same for 4a. Would it not be possible to show the number of negative vs. positive correlations between all pairs of nodes as, for example, a bar chart?
Table1: “Phopspholipids” and “Phospho-lipids”. Neither is correct. Please fix both.
Figure 5: Explain abbreviations used in the figure in the figure caption.
Round 2
Reviewer 1 Report
The authors have addressed all my comments and the updated manuscript is a much improved version of the one originally submitted.
Reviewer 2 Report
The authors have addressed all my previous comments and have much improved the manuscript. I still believe that the figures could be modified to improve reader comprehension and interest, but I also understand that not everything is possible.